# Multiobjective Optimization of Composite Wind Turbine Blade

**DOI:** 10.3390/ma15134649

**Published:** 2022-07-01

**Authors:** Mariola Jureczko, Maciej Mrówka

**Affiliations:** Department of Theoretical and Applied Mechanics, Faculty of Mechanical Engineering, Silesian University of Technology, Konarskiego 18A, 44-100 Gliwice, Poland; maciej.mrowka@polsl.pl

**Keywords:** composite, reinforcing materials, multiobjective optimization, wind turbine blade

## Abstract

When designing a wind turbine, the main objective is to generate maximum effective power with the lowest possible production costs. The power of a wind turbine depends primarily on the aerodynamic properties of its blades. Moreover, the cost of making a blade for a wind turbine, and therefore also for the entire wind turbine, depends on the materials used for its production. Therefore, wind turbine blades are the most studied element of a wind turbine. By selecting the optimal material and geometric properties of the wind turbine blade, it is possible to reduce the costs of making the entire wind turbine. These rationales led the authors to investigate composite wind turbine blades. A two-criteria optimization task was formulated, which allowed for the simultaneous consideration of two criteria: minimizing the mass and minimizing the vertical deflection of the wind turbine blade. Geometric properties of the blade, influencing the considered criteria, were assumed as decision variables. The weighted sum method was used. The results obtained allowed us to determine the optimal geometric and material properties of a wind turbine blade.

## 1. Introduction

The combination of the design process of a wind turbine blade with the optimization process allows reducing the costs of its manufacture. However, it is not an easy task. It requires taking into account many aspects that are incomparable, incalculable, and self-contradictory, which excludes their simultaneous optimization.

One of the basic criteria that the wind turbine should fulfill is that it produces the highest possible power at the lowest possible production costs. According to Ref. [1], the production of electricity from wind farms is unpredictable because it depends on changing weather conditions, therefore the total costs of wind energy generation should also include the costs of its storage. As the value of the output power of the wind turbine increases threefold to the blade length, the production of lighter and increasingly longer blades is profitable, and their appropriate selection is particularly important [2]. The cost of producing blades is only about 10% of the total cost of a wind turbine, so expenditure on innovation in blade designs, methods of manufacturing, and materials used are a relatively small share in the total cost of production. However, the lighter and better structured blade allows us to reduce the requirements for the hub and tower, thus reducing the production and operating costs of the entire wind turbine. In the design process of wind blades, an important task is to reduce their weight, which in turn contributes to reducing the weight and inertia loads of the blade. While the diameter of the wind wheel, and thus the “sweeping” area of the wind, increases in proportion to the square of the blade length, its mass, as a rule, based on experience, should increase threefold. In practice, however, this dependence has been mitigated by changes in the design form of blades and the development of methods of their production, optimizing the properties of composite structures from which the blades are made. The weight of the wind turbine blade can be reduced by changing its geometrical and material properties. NREL has developed a System Cost Breakdown Structure (SCBS) for wind turbines that provides the ability to view wind turbine components with varying degrees of cost detail [3]. Goeij W. et al. [4] present various concepts for the design of wind turbine blades, considering, inter alia, fatigue of composite structures loaded off-axis due to fiber orientation angles. This publication also contains descriptions of blade structures, techniques for their products, and materials from which they are made. In Ref. [5], the structural strength of large offshore wind turbine blades made of fiber-reinforced composites and with a spar in the form of a load-carrying box was examined. At the same time, the main goal of the research presented in [6], consisting of combining the structural and material model, was to better understand the progression of destruction in wind turbine blades.

The modern industry is constantly looking for construction materials that can provide improved functional properties and increased durability, substituting the engineering materials used until now. In Ref. [7], the authors showed the lack of linear dependencies between the properties examined, such as mass share and grain size, roughness parameters, and selected mechanical properties and the composition of the polymer composite with a matrix base of epoxy resin L285 cured with H285 hardener, and a physical modifier of friction in the form of alundum. The authors concluded that alundum of grain sizes equal to F220 and F240 exerted the best influence on the reduction of abrasive wear of the tested samples. However, in Ref. [8], the authors showed that chemical modification of PEEK samples resulted in deterioration of their thermoplastic properties. Moreover, the properties of the specimens manufactured using the 3D printing technology showed repeatability, dependent upon the method, of their processing parameters. The effects of printing direction of layers and structural form on the mechanical properties were observed. In Ref. [9], the synthesis and characteristics of functional polyesters with potential anticancer activity were described, followed by a post-modification process of biologically active polymers. Currently, a lot of interest in the manufacturing industry of construction materials is concentrated on the possibility of using composite. Composite materials are used due to their high-quality, low weight density, the possibility of processing with various methods, and the possibility of modifying properties by changing intermediate materials and processing methods The results in Ref. [10] showed that the introduction of fillers in the form of manganese residue and manganese (II) oxide fillers to an epoxy resin reduced the abrasive wear of the epoxy composites and simultaneously decreased the strength of the tested materials. According to the authors, the compositions containing the manganese (II) oxide filler were characterized by better mechanical properties than those containing the manganese residue filler. Researchers in Ref. [11] investigated the impact of the introduction of powder modifier into composite reinforced with carbon fabric on selected mechanical properties. They proved that an increase of alumina content by weight caused a decrease of the strength of polymer composites. The addition of alumina impeded the super-saturation of the carbon fabric, which caused discontinuities in the matrix material. The chapter in Ref. [12] is an introduction to high-density polyethylene (PE-HD) flame retarded compounds with 40–60% fraction of magnesium hydroxide (MDH). The authors described, inter alia, research to investigate the influence of surface modification of MDH and coupling agent on the flammability and mechanical properties of PE-HD/MDH composites. Moreover, the increased use of composite materials is caused by the possibility of changing their structural properties by adding modifiers to their composition. These modifiers and reinforcing fibers improve the material properties such as density, hardness, impact strength, mechanical and tribological properties, and reduced flammability [13,14,15,16,17]. The results of the research described in Ref. [13] showed an impact of the amount of added mass share modifier on the dynamical behavior of composite structures. According to the authors, the modifiers allowed one to change the stiffness of the structure without modifying the geometrical shapes or sizes. In Ref. [14], the authors investigated the processing-structure-property relationship for polymer blends. The described results indicated that there was an impact of the amount of added modifier on the dynamical behavior of the samples. Increasing the mass share of the physical modifier resulted in the increase of the material’s compliance. In another study [15], research was carried out to determine the effect of a selected physical modifier with different granularity and the mass percentage that was a component of the tested material on the vibration behavior of composite beams in the frequency range covering the first three forms of vibration. The authors showed that the dynamics of composite beams varied with the proportion of modifier. The research described in Ref. [16] showed that the use of an aerogel additive deteriorated the mechanical properties of the composite in relation to the base material (without the additive). It was also shown that the post-curing temperature changes the mechanical properties of the composite. In contrast, the aim of the research described in Ref. [17] was to examine the impact of weathering and thermal shocks on the abrasive wear of epoxy resin composites reinforced with carbon fabric that are used in aviation. However, the dependence of tribological resistance of the epoxy carbon-reinforced composite upon its hardness has not been demonstrated unequivocally. The weight of the wind turbine blades can be reduced by modifying the materials used for its structural elements, that is, supporting and stiffening the ribs and the sheathing. These materials usually contain fiber-reinforced materials such as fiberglass or carbon fibers and plastic polymers such as polyester resin or epoxy resin, and sandwich core materials which may be polyvinyl chloride (PVC), polyethene terephthalate (PET), or balsa wood [18]. Fiber-reinforced dying polymer composites are characterized by higher strength and lower weight. The most widely used are polymer matrix composites (PMC), metal matrix composites (MMC), and ceramic matrix composites (CMC) [19]. Modifications of composite materials from which the wind turbine blade shell is made may consist of increasing the thickness of the balsa layer and/or reinforcement layers, reducing the density of the balsa, and changing the orientation fibers or their type. The composites used for the stiffeners can be changed similarly. Many articles cover advanced materials used in the construction of wind turbine blades. Wind turbine blades are made of either carbon fiber or glass fiber composite material [20]. However, hybrid glass-carbon FRP is often used in large wind turbines due to similar mechanical properties, but at a lower cost and lighter weight compared to carbon or glass-fiber-reinforced FRPs. However, according to Ref. [20], defects can be generated in FRPs due to improper material selection during design or mis-operation during the layer-up process. One paper [21] described the materials that are commonly considered for the manufacture of wind turbines and the process used to recycle them. In this article, we present the scale of recycling methods for fiberglass and thermoplastic. According to Ref. [22], it is necessary to change the selection of materials used for wind turbine blades for materials that can be recycled or reused. The authors consider that the use of a polymer in a composite material as a blade material can lead to greater recyclability. In contrast, the resins used in the blades are typically thermosetting resins that form highly cross-linked polymers during the curing process, challenging recycling. In Ref. [23], the authors propose thermoplastic resins that can be heated and remolded as an alternative material for wind turbine blades that would facilitate recycling. A review of the composite materials used in the production of wind turbine blades is presented in Ref. [24]. In their article, the authors also presented the requirements for these materials due to the load condition of wind turbine blades. Traditionally used composites (glass fibers/composites with epoxy matrix), as well as natural, hybrid, and nanoengineering composites, are discussed. Their advantages and disadvantages are presented. On the other hand, Ref. [25] discusses the advantages of using thermoplastic composites due to their recyclability.

However, when designing wind turbine blades, it should be remembered that they should have a shape ensuring appropriate aerodynamic properties and adequate stiffness so that the blade does not collide with the tower at high wind speeds. The durability and life cycle of the wind turbine are a full 20 years. The designed wind turbine blade must also meet the requirement of a low level of generated noise. Wind turbine blade vibration is an undesirable phenomenon, as it contributes to the creation of additional dynamic loads and generates noise. Therefore, when designing wind turbine blades, they should be sought [26]. The outer shell of the blade ought to be resistant to different dirt and cooling, i.e., it must ensure high aerodynamic efficiency in various weather conditions.

Taking into account the above considerations, it can be noted that in the design and construction process of wind turbine blades, many aspects should be simultaneously considered, such as maximum strength, ensuring the stability of the structure, the minimum cost of execution, and the maximum value of generated power or minimum values of generated vibrations. There are many papers that describe the process of designing wind turbines in the literature [27,28,29,30]. The book in [27] is an overview of the recent advances and challenges encountered in wind turbine blade materials and design. It discusses the challenges and developments in wind turbine blade design and characteristics of wind turbine blade airfoils and the fatigue behavior of composite wind turbine blades. The advances in wind turbine blade materials, including biobased composites, surface protection and coatings, structural performance testing, and the design, manufacture, and testing of small wind turbine blades are also shown. The book in [28] is an invaluable reference for wind power engineers and researchers examining 3D printed miniature and large-size Savonious wind harvesters. Each chapter contains a detailed analysis and numerical and experimental case studies that consider recent research design, developments, and their application in practice. In Ref. [29], the authors described the general background for the design of wind turbine blades identifying the trends and issues of importance for these structures as well as concepts for “smarter” blades that address these issues. The topic of damage in composite materials and structural failures in the blades and the potential for blade design tailored to twist upon deflection are discussed. In Ref. [30], the authors showed a bionic design for wind turbine blades inspired by the airfoil of owl wings and the herringbone groove structure of owl feathers. The design of the bionic coupling blade is based on the bionic airfoil blade, which is coupled with the herringbone groove structure. The results presented indicate the power coefficient of the bionic airfoil blades is higher than that of the prototype blade. According to the authors, the herringbone groove structure improves the flow attachment by generating vortices, which reduce the pressure on the leeward surface of bionic coupling blades. However, although their design requires considering many criteria, most publications describe the optimal design of wind turbines only because of one criterion [31,32,33,34]. In Ref. [31], authors showed that multifidelity optimization methods are effective for a variety of wind energy applications to decrease the computational cost needed to find an optimal design. The authors used a traditional trust-region approach for multifidelity optimization but applied a new corrective function technique based on efficient KPLS surrogate models. They demonstrated the efficacy of this approach in three case studies. In Ref. [32], the authors used a genetic algorithm to determine the optimal shape of a wind turbine blade, ensuring a quick start-up of a wind turbine and its high output power while meeting only the strength criteria. As design variables were assumed, the chord, twist, and shell thickness along the blade were assumed. Obtained results showed that a hollow blade expedites the starting at low speeds by decreasing the blade inertia while the resultant stress along the blade does not exceed the allowable stress. On the other hand, in Ref. [33], the authors analyzed common goals used in wind turbine optimization problems. Maximizing annual energy production is shown to be significantly suboptimal compared to using integrated aero/structural metrics. In addition, minimizing the ratio of turbine mass to annual energy production is shown to be effective for fixed-rotor diameter designs if the tower mass is estimated carefully. Chen et al., in the paper [34], optimize wind turbine layout with many heights of the wind turbine hub using a greedy algorithm. The authors suggested that increasing the height of the turbine hubs inside the wind farm would also increase the total power output of commercial wind farms. There are far fewer publications that describe the problem of the multiobjective design of wind turbines [35,36,37,38,39,40]. In the paper [36], the authors used a multiobjective genetic algorithm (MOGA) to increase the productivity of the wind lens by designing the casing and turbine, as well as the height of the flange. The optimization process maximized power while minimizing drag and thrust. One article [37] proposed a multiobjective approach to making decisions when selecting a wind turbine problem using the weighted sum approach. A multiobjective optimization method for the structural design of horizontal-axis wind turbine blades was described in Ref. [39]. Cai et al. in their article [40] optimize the horizontal-axis wind turbine blades using the particle swarm optimization algorithm (PSO), which is joined with the finite element method (FEM).

In the article in [41], the authors discussed many optimization algorithms inspired by nature. Then, they compared terms of speed, accuracy, performance, convergence, efficiency, and complexity, concluding that “soft data processing techniques” allow for better results concerning “pattern-based algorithms”. Genetic algorithms have been shown to be characterized by fast performance, the best accuracy, and fast speed compared to other bio-inspired optimization techniques. On the other hand, in Ref. [42], the authors discuss 11 bio-inspired optimization algorithms; however, genetic algorithms are not taken into account. In contrast to the indirectly described review, the authors did not compare these methods with each other, nor did they define their advantages and disadvantages. On the other hand, in paper [43], the authors carried out a critical review of ten bio-inspired optimization techniques, more precisely three trajectory-based algorithms (the Simulated Annealing, the Great Deluge, and the Hill Climbing) and seven population-based (the PSO, ACO, ABC, BCO, BBO, ABO, and the GA). The authors indicated the dependence of the results obtained with the use of these optimization algorithms on the number of design variables. They stated that there is no guarantee that the same methodology used to obtain results in small problems will produce the same results in larger real-life environments. The paper in [44] provides a review of ten bio-inspired algorithms (the ANN, GA, ES, ACO, ABC, FSA, IWO, BBO, NSGA II, and the P-ACO). The authors did not demonstrate the advantage of any of the above-mentioned methods over the others. They only discussed the algorithms of their operation without pointing to any limitations in their use. They considered genetic algorithms to be the most used and therefore the most proven by researchers.

Based on the review of the current state of knowledge regarding the optimal design of wind turbine blades, the authors adopted as the material used for the wind turbine blade a traditional composite material, described in detail in Section 3. A genetic algorithm was selected to carry out the optimization process. The authors intend in the future to validate the obtained results, to optimize with other by bio-inspired methods. Due to the above considerations, a multiobjective optimization was conducted, during which both continuous variables (e.g., the thickness of longitudinal and vertical stiffeners) and discrete (e.g., the number of ribs and their distribution along the blade span) were considered. The optimization problem is formulated as the problem of discrete-continuous multiobjective optimization.

## 2. Assumptions Made during the Modeling of the Wind Turbine Blade

When designing the 27 m long blade for use in three-blade power plants with a horizontal axis of rotation, controlled by the angles of the blades, different aerodynamic profiles were used, thicker right at the rotor hub and thinner going towards the tip of the blade. The aerodynamic profiles FFA-W3 and RISØ series were used in the geometric model of the blade. The rated power of the analyzed wind turbine was 1 MW. The global tip ratio equal to 6 was assumed. The methodology of proceeding from adopting design assumptions, through determining the load condition and blade geometry, to carrying out the optimization process combined with numerical strength and dynamic analyses is presented in Figure 1.

The module for determining the load state of the blade using the modified Blade Element Momentum (BEM) method [2] and the module for analyzing the influence of the blade deformation on the aerodynamic load were prepared in MATLAB^®^ by MathWorks licensed from the Silesian University of Technology in Poland. The methodology to determine the geometry of a wind turbine blade and its state load is discussed in detail in [37].

The wind turbine blades have a tapering just at the connection to the hub of the wind turbine to reduce stresses in the rotor of the wind turbine. Directly at the rotor hub of the wind turbine, round aerodynamic profiles are used with a shell thickness to chord ratio equal to 100%, gently decreasing to a value of 14% at the tip of the blade. In the considerations presented, the blade was divided into 23 parts, creating 24 aerodynamic sections. Each section has a different aerodynamic profile. The most important geometric features of the investigated blade are shown in Table 1.

The CAD model of a wind turbine blade is shown in Figure 2. However, the FEM model of a wind turbine blade was created in Ansys^®^. Three groups of components were selected for this model:shells (upper and lower surface);ribs (transverse stiffeners);shear webs (longitudinal stiffeners).

By selecting elements in the numerical model of the blade, it is possible to assign them different thicknesses and material data, as well as define several types of elements. Furthermore, it is possible to determine the stresses, deformations, and masses of individual elements. The created FE model of the wind turbine blade consists of 12,343 elements. The 8-node shell99 was adopted as a finite element which has 6 degrees of freedom. This enabled the specification of any average or corner layer thickness at each node of the selected blade element, the direction angles of the layer material, and orthotropic material properties for any layer. This allows for modeling a composite material with a variable number of layers.

The shear webs are not twisted analogously to aerofoils. The shape of the blade with twisting the shear webs is shown in Figure 3.

## 3. Materials and Methods

To develop a numerical model of a wind turbine blade, a batch parametric file was developed in APDL for the Ansys program. In this way, it was possible to modify the numerical model of the blade at each iteration of the optimization process. The selection of three structural elements in this model made it possible to assign different thicknesses and material data, which are discussed in more detail in Section 2. The thickness of the stiffening ribs, the thickness of the shear webs, and the number of stiffening ribs and their location along the blade span are the variables entered into Ansys from the proprietary program in which the optimization process was carried out.

In their review, the authors discussed both traditional and modern materials used in the production of wind turbine blades. However, due to the lack of publication on the mechanical and physical properties of modern materials used for wind turbine blades, the authors used traditional composites for this purpose in their research. The choice was also dictated by the requirements of the cooperating company.

Tita et al. in their publication [45] described a method for making an orthotropic composite material, the matrix of which was epoxy resin and the reinforcement of glass fiber in the form of a bi-directional fabric. In this publication, the authors provided the material properties of the composite material obtained. The authors of this article used them to determine the material properties of shear webs and stiffening ribs.

The thickness of the transverse and longitudinal stiffeners is constant along the blade span. Using data published in Ref. [46], it was assumed that the shell of the wind turbine blade is a shell made of seven layers of material. The material properties of individual layers were taken from the same article. By slightly modifying the data given in Ref. [46], the following thicknesses of the individual layers were assumed: the first layer was a gel coat with a thickness of 0.51 mm; the second layer was a random mat (name taken from [46]) with a thickness of 0.38 mm; the third layer was a triaxial fabric (marked by the authors in [46] as A260) with a thickness of 1.27 mm; the fourth layer was balsa with a thickness of 0.75% length chord; the fifth layer was composed of alternating layers of triaxial fabric (marked by the authors in [46] as CDB340) and uniaxial (A260) fabric (where data of this layer are given as data for the spar cap mixture) with a thickness of 1% the airfoil thickness to chord ratio; the sixth layer was balsa with a thickness of 1.5% length chord; the seventh layer was 1.27 mm thick triaxial fabric (CDB340).

## 4. Formulation of Multiobjective Optimization Problem

The optimal design of the wind turbine blades is a very complex process. It requires considering many aspects that are incomparable, incalculable, and self-contradictory, which excludes their synchronized optimization.

The following requirements in the optimization of wind turbine blades design were considered by the authors:Minimization of frequencies of the blade vibrations.Maximization of output power.Minimization of material cost.Safeguarding stability of the blade.Fulfilment strength requirements.

Minimizing the frequency of blade vibrations is an effective approach to optimizing the design of wind turbine blades, at the same time optimizing other important aspects such as reducing manufacturing costs and increasing stability. However, when minimizing the frequency of blade vibrations, the natural frequencies of the blade should be separated from the harmonic vibrations related to the rotation of the rotor. In this way, the occurrence of resonance is prevented, which, with a large amplitude of vibrations, could lead to the destruction of the structure not only of the blade but of the entire wind turbine. Separation of natural frequencies from the resonance frequencies is one method of isolating frequencies.

The amplitude of vibrations generated by the wind turbine blade depends on its stiffness. The stiffness depends on material density, the thickness of the shell and shear webs, the thickness of the ribs, their number, and their arrangement along the blade span. Therefore, when the vibration minimization criterion is considered, the wind turbine blade should be offered with the highest possible strength.

Such an approach to the optimization of a wind turbine composite blade also allows for meeting another requirement, which is the generated maximum output power. The power generated by the wind turbine depends on the aerodynamic properties of the blade related to its geometry.

The cost of making a wind turbine blade depends on the same parameters such as the frequency of its vibrations. Thus, by formulating the optimization task as the task of minimizing the costs of making a blade, the task of minimizing its mass would be solved. However, such an approach would not ensure adequate blade stability. Its maximum stability is related to its maximum weight.

It is known that the weight of a wind turbine blade increases with use. This is related to atmospheric precipitation and deposits such as pollution, icing, glaze, and dust. A comprehensive analysis of consideration, the function relating to the minimization of the mass and the function relating to minimizing of the tip blade displacement, together with each having its weighting factor, so that the scalar function **F**(**X**) shown in Equation (1) now becomes a weighted sum the causes of possible damage to wind turbine blades related to precipitation and atmospheric sediments is presented in [21,47]. According to the Polish Wind Energy Association, in Polish climatic conditions, there is no need to use special technologies or materials to prevent the icing of wind turbines, which takes place in harsher climates. In Poland, when the turbine is iced, it is turned off. The shutdown period is so short that the impact on production efficiency is negligible. Taking the above into account, the authors of the article did not consider the changes in the mass of the wind turbine blade related to precipitation and atmospheric sediments in their research.

However, to meet the strength requirements of the wind turbine blade structure, the optimization task should be formulated as the maximization of its transverse displacement.

Given the above considerations, the authors formulated the optimization task as a multiobjective optimization task, which allows simultaneous analysis of quite a few objectives. The values of a particular objective depend on continuous parameters (thickness of shell, thickness of shear webs, and thickness of ribs) and discrete parameters (number of ribs, their arrangement along blade span).

The optimization task was formulated as a discrete-continuous multiobjective optimization task.

The multiobjective optimization task was solved using the weighted sum procedure, which transforms a vector function into a scalar function that is then minimized [48]. The solutions to this method are a set of Pareto-optimal solutions. This method is based on adding all the objective functions, in the case under of the objective functions and thereby transforms into a new optimization problem:(1) X∈Ωmin F(X)=wi·M+(1−wi)·D
where:Ω—the domain of possible solutions,**X**—column matrix of design variables,*w*—column array weights of the respective objective functions, which meet the assumptions $0\leq w_{i}\leq 1$,$M=\frac{m}{m_{per}}$—standardized objective function corresponds to the mass of the blade,*m*—the current mass of the blade,$m_{per}$—the permissible mass of the blade,$D=\frac{d_{tip}}{d_{tip\_per}}$—standardized objective function corresponds to the transverse displacement of the tip blade,$d_{tip}$—the current transverse displacement of the tip blade,$d_{tip\_per}$—the permissible transverse displacement of the tip blade.

The weighting factors represent the relative importance of individual objective functions for the decision-maker and are most often summed up to 1 in each optimization process. To generate the Pareto limit, a series of optimization steps were carried out covering the entire set of possible solutions, looking for the optimal solution for ω in the range 0 ≤ ω ≤ 1.

The set of possible solutions was determined by the boundary conditions imposed on individual design variables and the constraint conditions representing the criteria under consideration.

Four design variables in total are defined, which can be represented in the following form:(2)$$X={\left[x_{1},x_{2},\;x_{3},x_{4}\right]}^{T}$$
where $x_{1}$ is the thickness of ribs, $x_{2}$ is the thickness of shear webs, $x_{3}$ is the number of stiffening ribs, and $x_{4}$ is the arrangement of ribs. It should be emphasized that the independent design variables $x_{1}$ and $x_{2}$ are related to dependent variables, i.e., the number of composite layers from which they are made.

The design variables should be satisfied with the following inequality form:(3)$$X_{j}^{L}\leq X_{j}\leq X_{j}^{U}$$
where $X_{j}^{L}$ is the lower bound of the design variables, and $X_{j}^{U}$ is the upper bound of the design variables.

The following criteria are considered:(1)The strength criterion: the stress generated in the blade cannot exceed the permissible stress, namely, fulfillment with suitable strength requirements of the structure. The values of the permissible stress $\sigma_{per}$ were determined based on the material properties of individual materials, composite layers, used for the construction of the wind turbine blade.(4)$$\sigma \left(X\right)\leq \sigma_{per}$$(2)The global stability criterion: the local displacement of the wind turbine blade cannot exceed the permissible transverse local displacement of the blade $d_{per}$-global stability must be confirmed:(5)$$d\left(X\right)\leq d_{per}$$(3)The local stability criterion: the displacement of the tip blade cannot exceed the permissible displacement of the tip blade $d_{tip\_per}$; local stability must be confirmed:(6)$$d_{tip}\left(X\right)\leq d_{tip\_per}$$
where according to [49] the tip deflection should not exceed 20% of the radius of the windwheel.(4)The mass of the wind turbine blade must not exceed the permissible mass of the blade $m_{per}$.
(7)$$m\left(X\right)\leq m_{per}$$(5)The blade deformation must be smaller than the allowable the blade deformation $\varepsilon_{per}$-compliance with the local stability conditions of the structure:(8)$$\varepsilon \left(X\right)\leq \varepsilon_{per}$$(6)The vibration criterion: estrangement of the natural frequencies of the wind turbine blade from the harmonic vibration associated with rotor rotation:
(9)$$0.8\cdot f_{h}\leq f_{nl}\leq 1.2\cdot f_{h}$$where $f_{nl}$ is the *l*-th natural frequencies of the blade, and $f_{h}$ is the frequency of harmonic vibration. According to Ref. [49], the natural frequencies of the blade were located outside the scope of the harmonic vibration associated with rotor rotation at ±12%.

## 5. Results and Conclusions

Optimization tasks were performed in the proprietary program in which the algorithm of a modified genetic algorithm (MGA) was implemented. This program was developed in the Delphi environment. The MGA parameters used in the optimization process are listed in Table 2.

The probabilities of genetic operations were selected based on the results of the research discussed in [50].

The lower and upper bounds imposed on the design variables, determined based on design requirements, are summarized in Table 3. By design variable: stiffening ribs arrangement, we mean the position of individual ribs concerning the blade origin. These ribs are numbered 1 to 109, starting with the hub, every 0.25 m.

However, the values of the constraint conditions are presented in Table 4.

As a result of the optimization processes, the curve shown in Figure 4 was obtained, where the points are indicated for w = 0.0, 0.5, …, 1.0. The curve is called the Pareto frontier.

As a result of optimization, only non-dominated solutions were obtained. Therefore, all solutions for w = 0.0, 0.5, …, 1 form a Pareto frontier.

The obtained values of the criterion function, described by the Equation (1), depending on the adopted weighting factors, are shown in Figure 5.

From the set of Pareto-optimal solutions, the best solution was determined for the weight factor equal to 0.5. This means that later in the article, the solution obtained with the same weight of both objective functions is analyzed.

The values of the decision variables for the considered optimal solution are presented in Table 5.

The resulting optimal value for the thickness of the ribs is close to the upper bound. However, the obtained optimal value for the thickness of the shear webs is close to the lower bound. The distribution of ribs along the blade span for the considered optimal Pareto solution is shown in the Figure 6.

The optimal number of ribs obtained was 14. Their distribution along the blade span was ‘compacted’ from 2/3 of its length. Near the tip blade, there were 3 ribs placed every 0.25 m. This distribution was influenced by the form of one of the objective functions, i.e., minimization of the transverse displacement of the tip blade.

The summary of the mechanical and modal properties of a wind turbine blade with design features obtained from the literature (before optimization-preliminary model) and obtained as a result of the optimization is shown in Table 6.

Comparing the mechanical and modal properties of the wind turbine blades obtained for the initial model and the model with geometric features determined in the optimization process, which are summarized in Table 6, the following conclusions can be made.

The total mass of the wind turbine blade obtained after optimization is 10.7% greater than before the optimization, which increases the cost of the material. The greater mass of the wind turbine blade contributes to an increase in its stiffness. As a result, the maximum displacements of both nodes in the middle of the blade span and nodes in the tip blade were reduced by 16.7% and 15.1%, respectively.The blade tip displacement for the considered Pareto-optimal solution does not exceed the limit value; compared to the value obtained for the initial model, this value was reduced by more than 15%. This is due to the increase in the mass of the wind turbine blade, which is related to the stiffness of the structure.As a result of the optimization process, a wind turbine blade structure was obtained which is only 1% higher in the value of the 1st natural frequency. On the other hand, the natural frequencies for the second and fourth form increased by 8.2%.The range of natural frequencies of the numerical model of a wind turbine blade with geometrical and material features obtained as a result of the conducted optimization process does not coincide with the range of resonance frequencies. This is due to an increase in the mass of the wind turbine blade.

Figure 7 and Figure 8 show the obtained results of numerical simulation of vibration displacements in the transverse direction determined for the FEM model of the blade for selected nodes of the blade model, i.e., middle nodes and nodes of the tip blade, respectively, before and after the optimization process, with the best Pareto-optimal solutions were selected for comparisons.

The shape of the curves shown in Figure 7 for nodes located in the middle of the blade span indicates that there is inharmonic motion. On the other hand, the curves shown in Figure 8 indicate that the tip blade is moving in a harmonic motion.

Figure 9 and Figure 10 show the obtained amplitude-frequency characteristics of the numerical model of the wind turbine blade for selected nodes of the blade model, i.e., middle and tip blade, respectively, before and after optimization, with the best Pareto-optimal solutions selected for comparisons.

When analyzing the amplitude-frequency characteristics presented in Figure 9 and Figure 10, we can see that the first natural frequency has a value lower than the lower limit of the harmonic frequency range. On the other hand, the second eigenfrequency reaches a value greater than the upper range of frequency of harmonic vibrations (see Table 6). Therefore, there is no resonance phenomenon.

It should be emphasized that Pareto-optimal solutions are a compromise between ensuring the appropriate stiffness of the blade and a slight change in its mass.

Reassuming the optimization performed, it can be stated that the application of genetic algorithms made it possible to effectively shape the dynamic characteristics of a wind turbine blade, resulting in a significant reduction of its vibration amplitudes.

The authors intend to analyze the developed model of a wind turbine blade in terms of various changes in their future research. We intend to consider the use of other composite materials to make a wind turbine blade. The developed model is a parametric model, which also makes it possible to take into account additional masses during the operation of wind turbine (formed as a result of atmospheric precipitation and deposits) to determine their contribution to vibration of the wind turbine blade.

The aim of the research presented in the article was to develop a computational algorithm and to prepare computer programs for multi-criteria discrete-continuous optimization in terms of limiting the amplitudes of blade vibrations. Unfortunately, the results of these studies have not been verified by experimental studies carried out on a real model of a wind turbine blade. However, the research described in the article shows that the presented calculation algorithm effectively helps in determining the optimal design features at the design stage and ensures proper dynamic properties of the designed system. This allowed for the generalization of the developed methodology and application to solve the problems of optimization of dynamic properties in middle-class engineering systems, such as optimization of home wind turbine blades or optimization of the chassis structure of a rally car.

## Figures and Tables

**Figure 1 materials-15-04649-f001:**
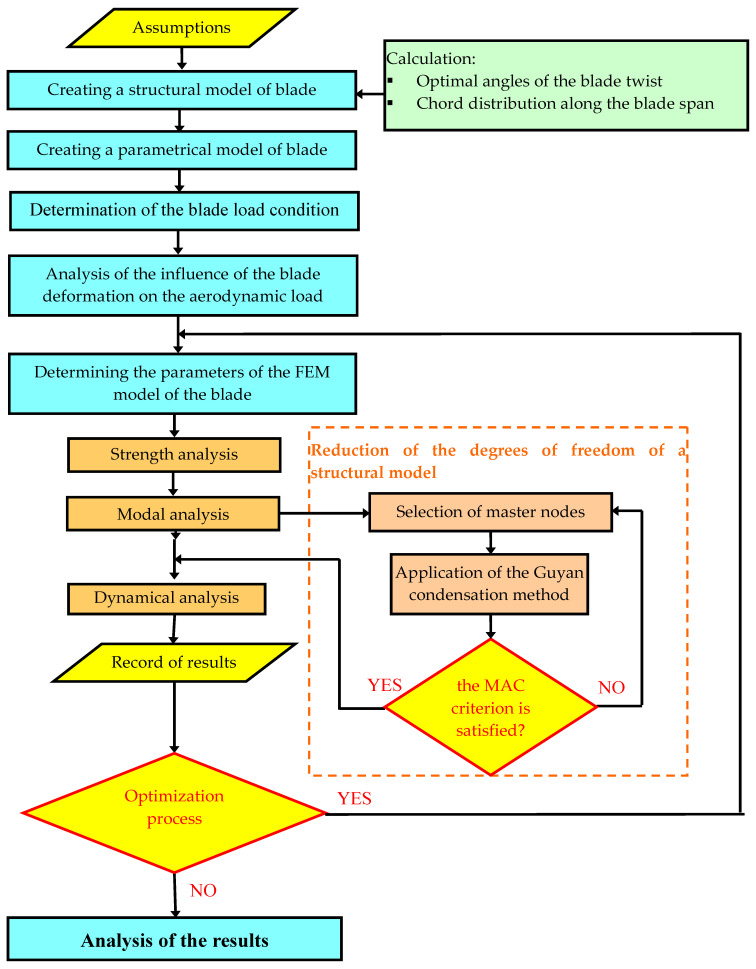
Flowchart of the numerical calculation algorithm.

**Figure 2 materials-15-04649-f002:**
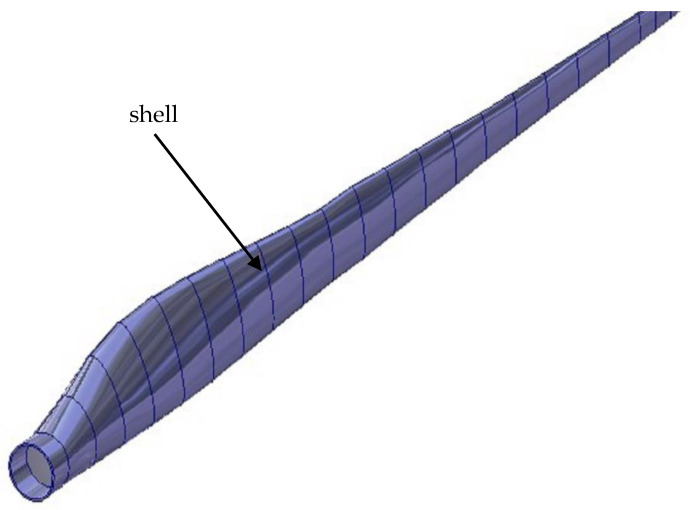
The model of the wind turbine blade in the CAD software.

**Figure 3 materials-15-04649-f003:**
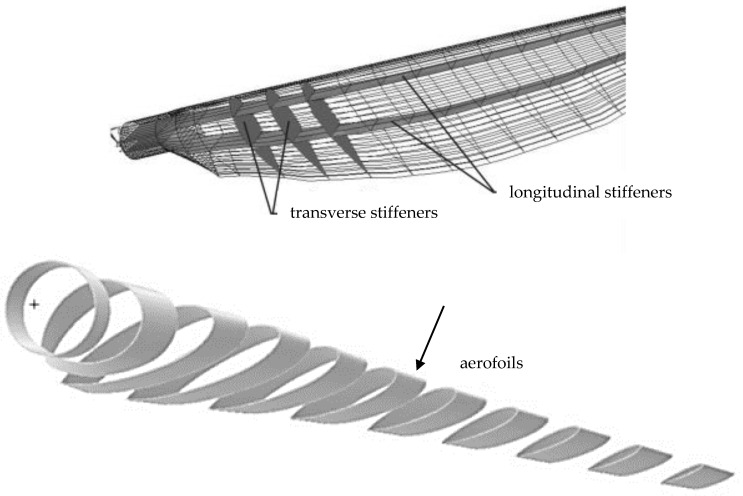
The numerical model of a wind turbine blade.

**Figure 4 materials-15-04649-f004:**
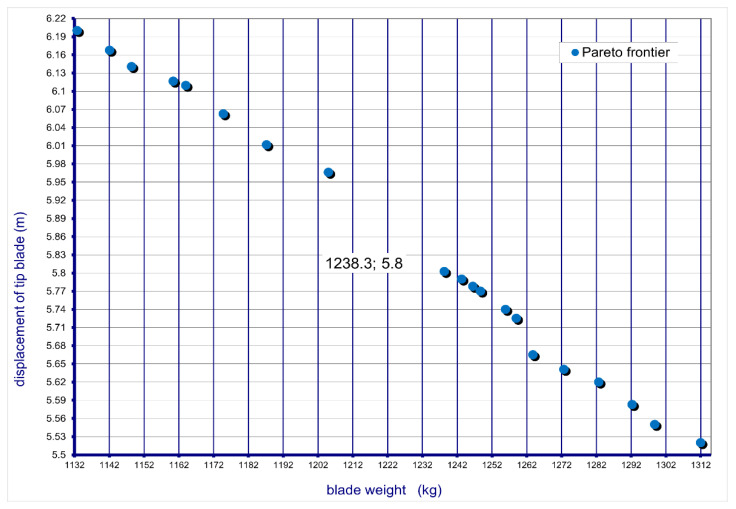
Pareto frontier for a multi-objective optimization problem with two objective functions.

**Figure 5 materials-15-04649-f005:**
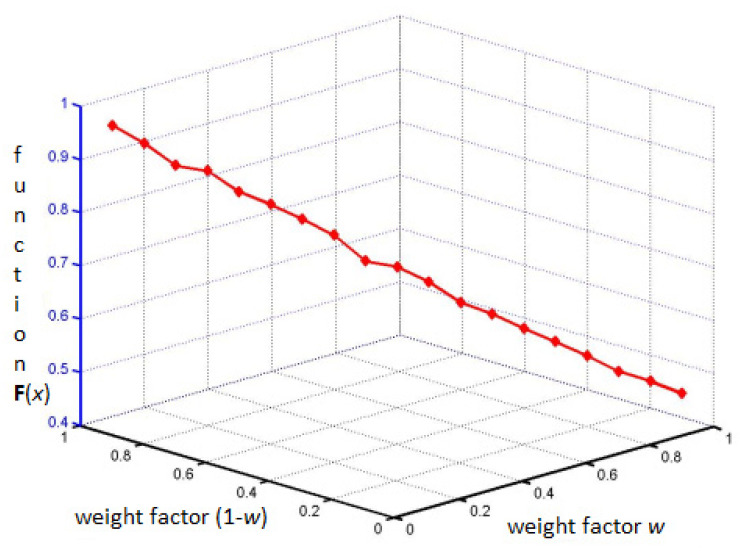
Graph presenting the values obtained for function (1) depending on the weight factors.

**Figure 6 materials-15-04649-f006:**
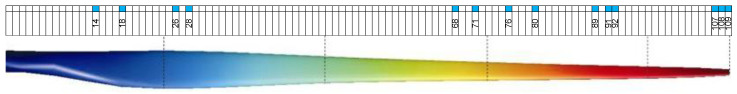
Graph presenting numbers of randomly selected ribs of the best individual for optimization process.

**Figure 7 materials-15-04649-f007:**
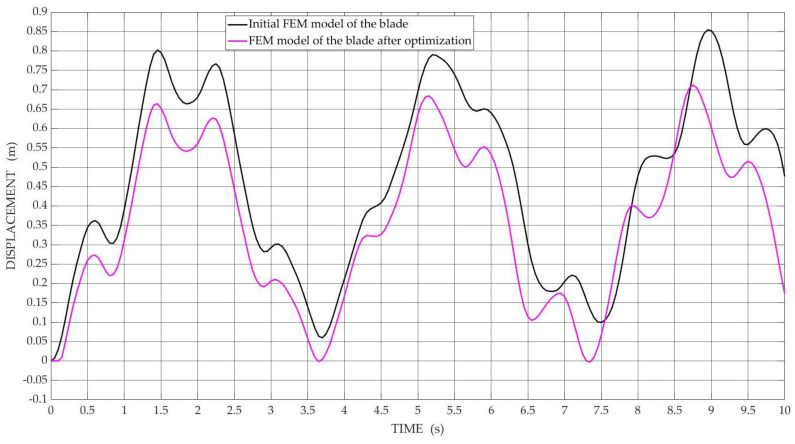
The diagram of vibration displacements in the transverse direction is determined in the node located in the middle of the blade span for the initial FEM model and the FEM model after optimization.

**Figure 8 materials-15-04649-f008:**
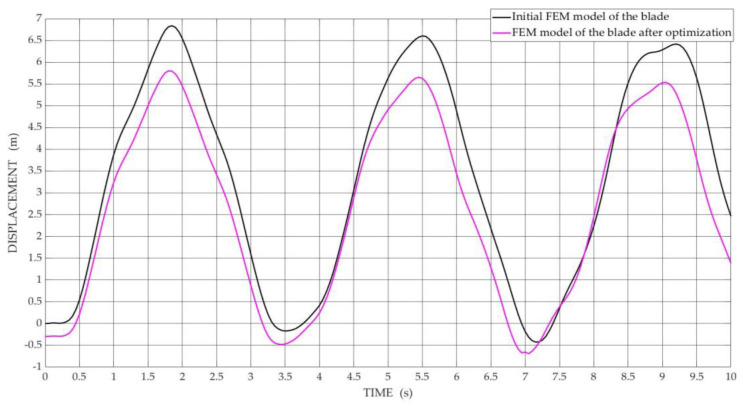
The diagram of vibration displacements in the transverse direction is determined in the node located at the tip of the blade span for the initial FEM model and the FEM model after optimization.

**Figure 9 materials-15-04649-f009:**
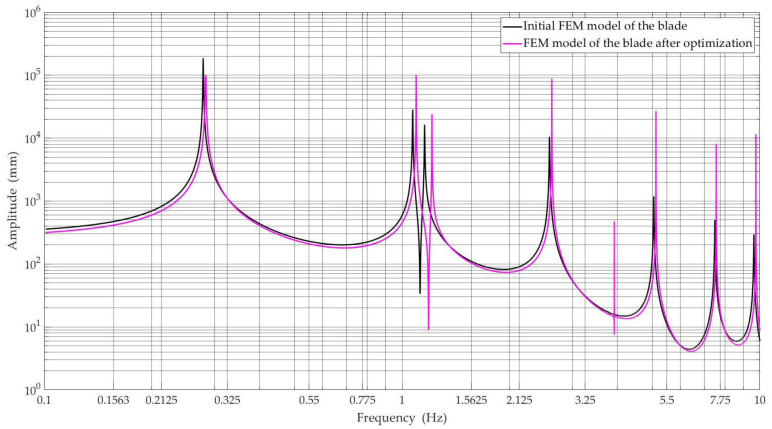
Amplitude-frequency characteristics were determined in the node located in the middle of the blade span for the initial FEM model and FEM model after optimization.

**Figure 10 materials-15-04649-f010:**
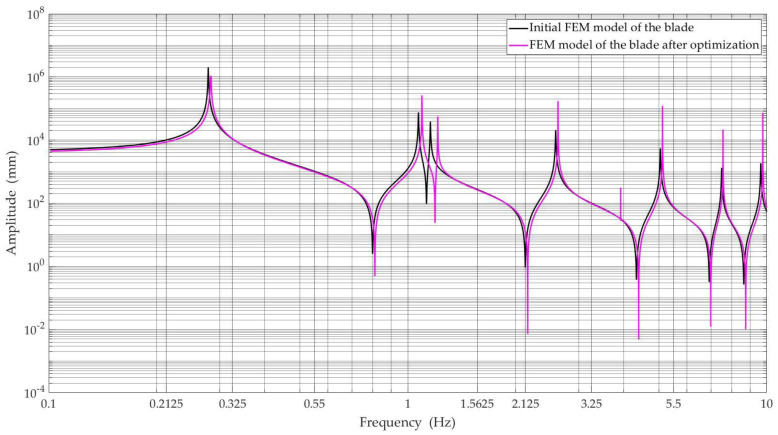
Amplitude-frequency characteristics were determined in the node located in the tip blade for the initial FEM model and FEM model after optimization.

**Table 1 materials-15-04649-t001:** Data on the physical characteristics of the wind turbine blade.

Data	Value	Unit
minimum chord	1	m
maximum chord	4	m
Minimum twist angle	0	°
Maximum twist angle	19.22	°

**Table 2 materials-15-04649-t002:** Implemented in the proprietary program MGA parameters.

Parameters	Value
Number of individuals	20
Number of populations (stop criterion)	50
Probability of crossover	0.7
Probability of mutation	0.3

**Table 3 materials-15-04649-t003:** The assumed in the proprietary program lower and upper bounds of the design variables.

Design Variable	Lower Bound	Upper Bound
The thickness of ribs, in (m)	0.01	0.10
The thickness of shear webs, in (m)	0.01	0.10
Number of ribs (-)	2	26
Arrangement of ribs, in (m)	0.25	26.75

**Table 4 materials-15-04649-t004:** The assumed in the proprietary program are the constraint conditions.

Parameter	Value	Unit
The permissible stress $\sigma_{per}$	375	MPa
The allowable deformation $\varepsilon_{per}$	0.5	%
The frequency of harmonic vibration $f_{h}$	0.3 ÷ 0.6	Hz
The permissible transverse displacement of the tip blade $d_{tip\_per}$	± 7	m
The permissible transverse local displacement of the blade $d_{per}$	± 7	m
The permissible mass of the blade $m_{per}$	2830	kg

**Table 5 materials-15-04649-t005:** The values of decision variables for the considered optimal solution.

Design Variable	Value
The thickness of ribs, in (m)	0.0963
The thickness of shear webs, in (m)	0.012
number of ribs (-)	14

**Table 6 materials-15-04649-t006:** The summarized properties of the wind turbine blade (International System—SI).

Parameter	Initial Model	Model after Optimization	Change
the maximum displacement of nodes in the middle of the blade span	0.8543	0.7120	↓ 16.7%
the maximum displacement of nodes in the tip blade	6.8355	5.8024	↓ 15.1%
maximum stress	227	204	↓ 10%
maximum strain (%)	0.4842	0.4428	↓ 8.5%
blade mass	1118.8	1238.3	↑ 10.7%
eigenfrequencies values			
1st mode shape	0.2784	0.2811	↑ 1%
2nd mode shape	0.973	1.053	↑ 8.2%
3rd mode shape	1.117	1.170	↑ 4.7%
4th mode shape	2.526	2.548	↑ 0.9%
5th mode shape	3.508	3.798	↑ 8.2%

## Data Availability

Not applicable.

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
