# Peer review of "Multiobjective Optimization of Composite Wind Turbine Blade"

_materials, 2022, doi:10.3390/ma15134649_

Round 1

Reviewer 1 Report

The authors investigated the optimization of composite wind turbine blade. The manuscript is found to be interesting after study it. However, its quality can be improved after addressing the following points: 

1. The manuscript should be checked for grammatical errors.

2. Abstract should be organized as the significance (two sentences), main problem, methodology and then some core outcomes of the study.

3. Some references added in bulk like [13-17], [21-24], [25-28] and so forth. Add at least two to three sentences about each study by following their major contribution.

4. which assumptions made during the study?

5. what are the practical applications of the conducted study?

6. The results should be validated with previously published literature.

7. The section 6 should be Conclusions instead of Discussion.  

Reviewer 2 Report

The authors presented a multi-objective optimization of wind turbine blades in regard to the frequency of blade vibration. The manuscript is well written and contains an analysis very useful in turbine design. Several details in regard to the design of the study require a better explanation as indicated in the attached pdf file.
Although a short literature review of algorithms and materials was presented, this is currently missing important blade materials, as well as a critical review of the optimization algorithms and their relation to the results of the manuscript. For these reasons, the manuscript cannot be published in its current form.

Reviewer 3 Report

I suggest the paper should not be accepted in this form.

1. The abstract should be rewritten, I do not know what is the main method or results in the abstract. 2. The literature should be updated, more literature should be in recent three years. 3. In the introduction, the disadvantages of the references should be summarized clearly to emphasize the importance of this work. 4. Part 3 should be rewritten, The initial and boundary conditions of the model should be given in details. The conditions could be listed in a table. 5. There are too many formulas and symbols, so a nomenclature table is needed. 6. I suggest that Part 5 and Part 6 be merged. The explanation of Fig.4-7 is too short. It should be explained in details. More explain should be given according to every result.

Round 2

Reviewer 2 Report

The authors applied significant revisions according to the reviews that greatly improved the manuscript. Where permitted, please consider adding all replies to comments in the manuscript for the benefit of the reader, e.g. rev1-f, rev1-g.

Reviewer 3 Report

It can be accepted